# Glioblastoma Multiforme—A Look at the Past and a Glance at the Future

**DOI:** 10.3390/pharmaceutics13071053

**Published:** 2021-07-09

**Authors:** Jasmine L. King, Soumya Rahima Benhabbour

**Affiliations:** 1Joint Department of Biomedical Engineering, North Carolina State University and The University of North Carolina at Chapel Hill, Chapel Hill, NC 27599, USA; jking116@live.unc.edu; 2Division of Pharmacoengineering and Molecular Pharmaceutics, UNC Eshelman School of Pharmacy, University of North Carolina at Chapel Hill, Chapel Hill, NC 27599, USA

**Keywords:** glioblastoma, blood–brain barrier (BBB), blood–brain tumor barrier (BBTB), surgery, radiation, chemotherapy, immune checkpoint inhibitors (ICIs), stem cell-based therapy, stem cell engineering, hydrogels, smart hydrogels, focused ultrasound (FUS)

## Abstract

Gliomas are the most common type of brain tumor that occur in adults and children. Glioblastoma multiforme (GBM) is the most common, aggressive form of brain cancer in adults and is universally fatal. The current standard-of-care options for GBM include surgical resection, radiotherapy, and concomitant and/or adjuvant chemotherapy. One of the major challenges that impedes success of chemotherapy is the presence of the blood–brain barrier (BBB). Because of the tightly regulated BBB, immune surveillance in the central nervous system (CNS) is poor, contributing to unregulated glioma cell growth. This review gives a comprehensive overview of the latest advances in treatment of GBM with emphasis on the significant advances in immunotherapy and novel therapeutic delivery strategies to enhance treatment for GBM.

## 1. Introduction

Gliomas are classified as the most common tumors of the brain and spinal cord that develop from glial cells in the central nervous system (CNS). In the CNS, glial cells consist of oligodendrocytes, astrocytes, and microglia [1,2]. These cells are non-neuronal and their main functions are to provide support and protection, and regulate hemostasis in the CNS. The specific origin of the glial cell determines the type of glioma formed. There are three different types of gliomas: ependymomas which arise from glial cells in the epithelial lining of the brain and spinal cord, oligodendrogliomas which originate from oligodendrocytes, and astrocytomas which develop from astrocytes [3,4]. Of those, astrocytomas are the most frequently occurring glioma in pediatric, adolescent, and adult patients. In adults, astrocytoma grade IV or glioblastoma multiforme (GBM) constitutes approximately 15.6% of brain tumors and 45.2% of primary malignant brain tumors [5].

GBM is the most aggressive, highly malignant tumor of the astrocytic lineage and is commonly diagnosed in elderly patients (median age at diagnosis ≥ 65 years) [6]. Glioblastomas are characterized by extensive, diffuse tumor invasion and infiltration, microvascular proliferation, and high genomic instability [7]. Cancer stem cells (CSCs) in glioblastoma contain tumorigenic properties that contribute to tumor progression, therapeutic resistance, and tumor recurrence [7]. CSCs contain driver mutations that promote intratumoral heterogeneity and aberration of signaling pathways [7]. This further promotes tumor survival, proliferation, and metastasis. Several key signaling pathways that are dysregulated in GBM are (1) the tumor protein p53 (p53) pathway, (2) the mitogen-activated protein kinase/extracellular signal-related kinase (MAPK/ERK) pathway, and the retinoblastoma protein pathway (RB) [8]. These molecular expression patterns have a major clinical significance that determines prognosis and response to therapy. Primary GBMs are genetically characterized by epidermal growth factor receptor (EGFR) amplification, phosphatase and tensin homolog (PTEN) mutation, and absence of isocitrate dehydrogenase (IDH) mutations [8]. TP53 mutations are the most frequent genetic alteration observed in secondary GBMs [8]. Primary GBMs are more common in elderly patients, whereas secondary GBMs develop in younger, adolescent patients [8]. The median survival for patients diagnosed with GBM is approximately 12 to 15 months [9]. Despite decades of research to improve patient outcomes, GBM still remains incurable and very challenging to treat.

In this review, we discuss the current management of adult GBM, promising immune checkpoint inhibitor therapies in clinical trials, and novel emerging therapeutic approaches that have potential to advance GBM treatment.

## 2. Current Treatment for Glioblastoma Multiforme (GBM)—*Where We Are Now?*

### 2.1. Surgical Resection

The current gold standard of care for GBM is surgical resection followed by adjuvant chemotherapy and radiotherapy. Given the poor prognosis of GBM, surgical debulking of the tumor mass is often performed to reduce tumor burden and improve survival benefit. Neurosurgeons must evaluate the tumor size and location, and patient’s functional status to determine the extent of resection (EOR) that prolongs patient overall survival (OS), improves quality of life, and preserves neurological function [10]. Neurosurgical options for GBM include biopsy, gross total resection (GTR), or subtotal resection (STR). GTR is defined as maximal removal of the tumor observed by magnetic resonance imaging (MRI). In contrast, STR is defined as removal of a portion of the tumor and residual tumor lesions are observed in post-operative images. Based on the systematic reviews and meta-analyses conducted by Brown et al. and Han et al., studies showed that GTR significantly improves progression-free survival (PFS) and overall survival for GBM patients in comparison to STR [11,12]. However, multiple tumor lesions, bilateral tumor involvement, and bulky tumors pose a clinical challenge and risk for total resection [13]. Due to clinical infeasibility, STR is used as an alternative operative approach. Although maximal surgical resection has been shown to improve patient overall survival and quality of life, recurrence is inevitable.

### 2.2. Radiotherapy

Radiotherapy (RT) is considered an adjunct therapy following surgical removal of GBM to target residual cancerous lesions in the resection cavity. RT uses high-energy beams to destroy cancerous cells by causing DNA damage, thus inhibiting cell cycle progression. In post-operative GBM patients ≤ 70 years of age, conventional fractionated RT is often prescribed at a conventional dose of 60 Gy in 2 Gy fractions over 6 weeks [14]. However, in elderly patients (≥70 years of age), hypofractionated short-course radiotherapy (SCRT) may be preferred over conventional radiation treatment. Hypofractionated SCRT delivers higher doses per fraction of radiation treatment over a shorter period of time [15]. Factors such as tumor size/location, metastasis, and Karnofsky Performance Status (KPS) play a significant role in patients’ prognosis. However, age is a major prognostic factor in GBM and often guides treatment decisions [15]. Elderly patients (≥70 years of age) are likely to have underlying comorbidities, concomitant diseases, and significantly more molecular alterations at diagnosis, which adds complexity to treatment recommendations. Additionally, there is no clear clinical consensus on proper management of this patient population due to their exclusion from clinical trials. Given these limitations, the National Comprehensive Cancer Network (NCCN) guidelines recommend hypofractionated RT in elderly patients (≥70 years of age) [16]. A prospective randomized controlled trial was conducted by Roa et al. to investigate the difference in overall survival outcomes in elderly GBM patients (≥60 years of age) undergoing conventional RT (60 Gy in 30 fractions over 6 weeks) vs. SCRT (40 Gy in 15 fractions over 3 weeks) [17]. The results of the study demonstrated non-inferiority in overall survival between patients receiving 40 Gy/15 fractions and 60 Gy/30 fractions (5.6 months vs. 5.1 months; *p* = 0.57), respectively. However, patients receiving conventional RT required an increase in post-treatment corticosteroid total daily dose in comparison to patients in SCRT group (49% vs. 10%; *p* = 0.02). Furthermore, RT was discontinued in fewer patients receiving SCRT (10%) than conventional RT (26%). Shorter courses of RT (25 Gy/5 fractions over 1 week) have been explored and results have shown that further reducing the treatment duration may be clinically appropriate for elderly and/or frail patients newly diagnosed with GBM [18]. Based on these studies, hypofractionated SCRT in elderly patients can reduce medical cost, post-treatment pill burden, increase the probability of RT completion, which can ultimately enhance quality of life.

Repeat radiation to treatment volumes using standard radiotherapy approaches can cause radiation induced neurotoxicity to healthy neuronal tissue resulting in neurocognitive dysfunction. New advances in imaging technologies have improved the delivery of radiation treatment and have been shown to be more precise and effective at targeting tumorous tissue. Intensity modulated radiation therapy with image guidance (IMRT/IGRT) uses computer-generated software to deliver, shape, and focus radiation doses on the target tumor tissue [19]. The advantage of combining image guidance allows for imaging prior to and during each radiation dose to improve delivery and accuracy of radiation treatment. This technological advancement optimizes the location, shape, and target dose of radiation, decreases dose to adjacent normal tissue volumes, and limits dose heterogeneity within the target volume [20,21]. While advances in radiological treatment of GBM are encouraging, this has not translated into improved survival and has not been shown to overcome radioresistance.

### 2.3. Chemotherapy

First-line adjuvant chemotherapy for newly diagnosed patients with GBM is temozolomide (TMZ). TMZ is a DNA-alkylating agent that exerts its cytotoxic effects by methylating the O^6^ position of guanine in DNA [22]. This causes a disruption in the DNA structure and induction of cell cycle arrest, which ultimately leads to apoptosis of cancer cells. The efficacy of TMZ in GBM patients is correlated to intracellular levels of O^6^-methylguanine-DNA methyltransferase (MGMT) protein. MGMT, a DNA protein, reverses the effects of alkylating agents by demethylating the O^6^ guanine residue, thereby reducing the sensitivity of TMZ to glioma cells. The desensitization of glioma cells to TMZ increases TMZ resistance, enhances tumor growth, proliferation, and infiltration. Given the ability of GBM cells to circumvent TMZ antitumor activity, this leads to treatment failure and reduced survival outcomes. To potentiate the effects of TMZ in GBM patients, studies have assessed the combination of TMZ with antiangiogenic agents, tumor treating fields, and immunotherapy [23,24,25,26,27].

Bevacizumab (BVZ; Avastin) was FDA approved as adjuvant therapy in patients with recurrent GBM in 2009. Bevacizumab, a vascular endothelial growth factor (VEGF) inhibitor, acts by binding to circulating VEGF to prevent ligand-receptor interaction at the cell surface [28]. This inhibition leads to the reduction in tumor vascularity and growth. A multicenter, randomized, open-label phase 3 EORTC 26101 study investigated the combination of bevacizumab (10 mg/kg every 2 weeks) plus lomustine (90 mg/m^2^ every 6 weeks) versus lomustine (110 mg/m^2^ every 6 weeks) alone in 432 patients with recurrent GBM [29]. The addition of BVZ to lomustine did not result in improved overall survival; however, median progression-free survival was extended with the addition of BVZ to lomustine versus lomustine alone (4.2 months vs. 1.5 months). In a randomized, double-blind, placebo-controlled phase 3 study (Avastin in Glioblastoma; AVAglio), the addition of bevacizumab (10 mg/kg every 2 weeks) to radiotherapy (2 Gy 5 days a week) and oral TMZ (75 mg/m^2^ for 6 weeks) was evaluated in newly diagnosed GBM patients to determine the effect on progression-free and overall survival. Following an initial treatment regime, maintenance therapy was continued for six 4-week cycles with bevacizumab (10 mg/kg every 2 weeks) or placebo, plus TMZ (150–200 mg/m^2^ for 5 days). Results of this trial did not significantly improve overall survival; however, the addition of bevacizumab prolonged median progression-free survival with respect to the placebo group (10.6 months vs. 6.2 months; *p* < 0.001). The combination of BVZ with other chemotherapeutics has been investigated and shown similar results in improving progression-free survival but there was no statistical improvement in overall survival. Although BVZ is generally well tolerated, reports have shown that patients are at risk of developing intracranial hemorrhages, thromboembolic events, and gastrointestinal perforation while on BVZ therapy, which can lead to discontinuation of therapy [30].

Carmustine wafer (BCNU; Gliadel) implants are recommended as adjunctive therapy for the treatment of newly diagnosed and recurrent GBM. The biodegradable wafers are implanted into the resection cavity to achieve controlled delivery of BCNU to glioma cells. Although this treatment approach bypasses systemic toxicities, there are several complications associated with the implantation of Carmustine wafers. Case studies have reported surgical site infections, extensive cerebral edema resulting in neurological deficits, pericavity necrosis, and severe hydrocephalus [31,32].

The current first-line, second-line, and salvage chemotherapy options for the management of GBM have prolonged overall survival and improved the quality of life in GBM patients when used in combination. However, to date, GBM remains incurable and is universally fatal.

## 3. Immune Checkpoint Inhibitors in GBM

The blood–brain barrier (BBB) is a highly selective semipermeable membrane that mediates the interaction and passage of materials from the periphery to the central nervous system (CNS). This protective barrier reduces the levels of immune cells circulating, thus limiting immune responses in the brain. However, tumors can compromise the integrity of the BBB, causing an increase in vascular permeability and extravasation of immune cells. To prevent immune attack, GBM tumor cells release tumor-associated antigens (TAA) that are taken up by resident macrophages and presented to T cells to suppress their immune effector function [33]. Furthermore, GBM tumor cells upregulate their expression of immune checkpoint proteins to potentiate immunosuppressive activity and GBM tumor cell evasion. Research efforts have focused on understanding these signaling mechanisms that allow cancer cells to evade the immune system, suppress T-cell function, and migrate to distant locations [27,34]. There are several active, ongoing clinical trials evaluating the efficacy of immune checkpoint inhibitors alone or in combination with standard of care for newly diagnosed GBM or recurrent GBM (Table 1). These studies are currently awaiting publication but preliminary data have been reported. In the CheckMate-548 clinical trial, the combination of nivolumab with first-line GBM therapy failed to meet PFS primary outcome measure and the investigators are currently awaiting for overall survival data (NCT02667587). Similarly, disappointing primary outcome measures were observed in the phase II study evaluating the safety and efficacy of atezolizumab in combination with first-line treatment (NCT03174197). However, reports showed that concurrent atezolizumab with TMZ and radiotherapy was well tolerated and no safety concerns were observed. GBM recurrence was observed in several patients post-atezolizumab treatment. Seventeen patients received repeat surgery and analysis of tumor tissue pre- and post-immunotherapy may provide clinical insight on immunotherapy resistance. A phase II study is evaluating the pharmacodynamic effects of pembrolizumab in newly diagnosed patients with GBM. Twenty participants have been enrolled in this study and currently there are no study updates on primary outcome measures (NCT02337686). Lastly, in a single-center, phase 2, open-label study, the combination of avelumab with standard therapy was shown to be safe and generally well tolerated in newly diagnosed GBM patients (NCT03047473). Although the efficacy data are premature and results are preliminary, avelumab may be promising in the initial stages of GBM therapy.

Previous clinical trials investigating the efficacy of immune checkpoint inhibitors in patients with GBM have also not been shown to meet primary outcome measures (NCT02617589) [35]. Although the endpoint analysis was not promising, there is hope that these previous and ongoing clinical trials can help identify better treatment approaches for patients with primary or recurrent GBM.

## 4. Clinical Need to Target Tumor Infiltration

The highly infiltrative nature of GBM poses a clinical challenge for conventional chemotherapeutics and targeted therapies. The molecular and genetic alterations within glioma cells contribute largely to its tumor heterogeneity, stemness, and invasiveness. Additionally, glioma cells recruit tumor-associated macrophages (TAMs) that provide protection and evasion from surrounding immune cells, which further promotes GBM infiltration, migration, and distant tumor involvement. GBM metastases can involve tumor expansion in contralateral hemispheres, brainstem, spine, leptomeninges, and extracranial sites such as lung, bone, lymph nodes, liver, soft tissue, and skin [36,37,38]. Extracranial GBM is rare and more frequently develops in younger patients due to better biomolecular profiles and survival outcomes [38]. Tumor expansion in contralateral hemispheres is most commonly observed in patients with GBM. These tumor cells evade the primary origin of GBM, migrate across the corpus collosum, and proliferate to form a new tumor lesion. The dysregulated molecular pathways that facilitate the propagation of glioma cells render localized therapy and conventional chemotherapeutics ineffective. There are few chemotherapeutics that are able to cross the BBB and accumulate in the tumor tissue at therapeutic concentrations. Given the complexity of the tumor microenvironment (TME) and infiltrative nature of GBM, dose escalations and various combination therapy approaches are often required to target the infiltrating growth of GBM. However, intensifying the course of therapy for infiltrated glioma cells is limited by dose-limiting toxicities and treatment-induced neurological deficits. It is important to note that survival outcomes and quality of life measures worsen from baseline with multifocal involvement and recurrence [39,40]. Therefore, better understanding of the TME and molecular pathways can provide a novel strategy to inhibiting glioma cell evasion and infiltration. Furthermore, the development of personalized therapy approaches that can inherently target infiltrative lesions in real time are warranted to overcome the limitations of current treatment strategies for GBM.

## 5. Clinical Need for Drug Delivery Systems

Conventional chemotherapy remains the standard therapy option for primary and recurrent GBM. Therapy management is tailored specifically to the individual patient based on prognostic biomarkers; however, chemotherapy options are limited due to BBB-associated delivery challenges, susceptibility to rapid systemic clearance and degradation, and dose-limiting toxicities. These issues highlight the need for drug delivery systems that can enhance drug bioavailability and half-life, improve drug penetration across the BBB, and promote drug distribution and accumulation in tumor tissue while minimizing systemic toxicities. In 2003, the polyanhydride biodegradable implant Gliadel^®^ containing carmustine was FDA approved for intracranial use as adjunct therapy in newly diagnosed and recurrent GBM [41]. This polymeric delivery system was designed to sustain the release of BCNU after surgical resection. Given the localized placement of the wafer to residual tumor cells, this approach conferred advantages over systemic administration of BCNU. When delivered systemically, BCNU is rapidly metabolized with a relatively short half-life and studies have shown limited clinical efficacy and severe hematological side effects following therapy [31,32,42,43]. The intermittent exposure of BCNU to tumor cells following intravenous administration is a shortcoming of systemic chemotherapy treatment which ultimately impacts survival rates. In contrast, the Gliadel wafer directly delivers BCNU to the tumor site, thus (1) enhancing drug distribution and accumulation in tumor tissue, (2) sustaining release of BCNU over weeks, and (3) immediately exposing residual tumor cells to therapy following surgery. Though this was the first polymeric implant for localized therapy in GBM, the use of Gliadel wafers was associated with severe neurological complications and infections after placement. Giladel^®^ is the only biodegradable, implantable polymeric system that has demonstrated the ability to impregnate and deliver a chemotherapeutic drug directly to the tumor. This emphasizes the need to focus on developing the next generation of drug delivery technologies that can further optimize drug delivery and improve treatment for GBM.

## 6. Clinical Need to Increase Delivery of Therapeutics across Blood–Brain Barrier (BBB)

The main function of the BBB is to protect and prevent foreign pathogens and toxins from affecting healthy brain tissue. Specialized endothelial cells that contain tight gap junctions are responsible for creating the border that regulates the passage of substances from the periphery into the CNS. Drug delivery to the brain relies heavily on the drug molecule’s ability to traverse the BBB. A major hurdle for most therapeutic compounds is their inability to freely transport across the BBB. Additionally, some drug molecules may passively diffuse through the BBB but drug concentrations within the target brain tissue are subtherapeutic. Smaller (<500 Da), lipophilic (log *P* 1.5–2.5) compounds are more favorable drug candidates for brain diseases due to their ability to permeate through the lipid bilayer of the BBB [44]. However, only 5% of small-molecule drugs enter the brain parenchyma to treat CNS diseases with the most commonly conventional chemotherapeutics for GBM provided in a summary table below (Table 2) [45]. Although these chemotherapeutics may freely pass the BBB, CNS endothelial cells express drug efflux transporters, mainly MDR1, that play a role in hampering drug accumulation within the brain parenchyma (Figure 1) [46].

In brain tumors, the integrity of the BBB is disrupted, but tumor neovascularization and angiogenesis stimulate the production of new blood vessels that are heterogenous and hyperpermeable in comparison to normal vasculature [47,48]. The remodeling of the tumor vasculature leads to the development of the blood–brain tumor barrier (BBTB). Given the enhanced permeability of the BBTB, the delivery of drugs to the brain tumor core can be greatly increased and retained, thus promoting a more efficient, therapeutic response. As mentioned, the tumor vasculature is heterogenous in nature and therefore some areas are more impermeable and resistant to drug delivery. Furthermore, nanomedicine technologies have been explored to overcome the BBTB in hopes of improving the delivery of chemotherapeutics to brain tumors. Nanotechnology based drug delivery is an emerging field that preferentially exploits the leaky tumor vasculature to achieve greater drug penetration, drug distribution, and drug retention within the tumor. This phenomenon is known as the enhanced permeability and retention (EPR) effect [49,50]. These nano-drug delivery systems can be designed to possess unique features that increase drug loading capacity, improve solubility of poorly soluble drugs, enhance drug stability, and allow for more targeted drug delivery via surface modifications [50,51,52]. There are two FDA-approved nanoparticle-based treatments for use in cancer—Doxil^®^, a liposomal formulation containing doxorubicin, and Abraxane^®^, an albumin-based nanoparticle formulation containing paclitaxel. Though nano-based drug delivery systems harness the EPR-mediated tumor targeting effect, the accumulation of nanocarriers within the tumor is highly variable, thus owing to unpredictable therapeutic outcomes between patients [53]. This highlights the opportunity for designing and developing new strategies to improve the delivery of therapeutics across the BBB/BTBB.

## 7. A Glance at the Future

### 7.1. Anticancer Stem Cell Therapy for GBM

Gene therapy has been extensively studied for the treatment of GBM. The clinical translation of viral delivery for GBM has been challenging due to inefficient tumor penetration and limited clinical efficacy. In recent years, investigative efforts have focused on genetically modifying stem cells (SCs) to produce antitumor agents. Stem cells display unique tumoritropic and immunosuppressive properties that make them attractive cell carriers and superior to traditional viral vector delivery. Aboody et al. were the first to discover that neural stem cells (NSCs) have an inherent ability to home to brain tumors [54]. In this preclinical study, implanted NSCs selectively migrated, co-localized with intracranial tumors, and delivered cytotoxic protein to suppress tumor growth. This study conferred that NSCs display an extensive tumor tropism for brain tumors. Other studies have shown that different stem cells, including mesenchymal stem cells (MSCs), induced pluripotent stem cells (iPSCs), and embryonic stem cells (ESCs), possess similar tumor tropic behavior and migratory properties [55,56,57,58]. There are several ways SCs have been genetically modified to attenuate tumor growth. Here, we describe SC engineering strategies that have shown great promise in treating GBM.

#### 7.1.1. Engineering Stem Cells to Secrete Anticancer Proteins

SCs can be engineered with therapeutic genes encoding secretable effector molecules that function to generate antitumor activity (Figure 2). Effector molecules that have been used to regulate tumor growth include tumor necrosis factor apoptosis-inducing ligand (TRAIL), interferons (α/β), interleukins (ILs), and single-chain antibodies [59,60]. The antitumor effects of SCs expressing TRAIL have been extensively studied in GBM [61,62,63,64,65,66,67,68]. TRAIL is a pro-apoptotic ligand that binds to death receptor (DR) 4 and 5 on cancer cells. Intracranial delivery of engineered neural stem cells expressing TRAIL (iNSC-sTR) were investigated by Bagò et al. to determine their potential as cellular delivery vehicles for GBM therapy. The results from this in vivo preclinical study demonstrated that iNSC-sTR therapy effectively suppressed tumor growth by 18.3-fold 33 days after treatment and extended median survival to 62 days in GBM8 tumor-bearing mice, with respect to control [66]. Buckley et al. demonstrated that autologous, patient-derived neural stem cells can be generated from skin fibroblasts and maintain tumor-homing properties while expressing high levels of TRAIL to suppress tumor growth (Figure 2) [67]. Though TRAIL stem cell-based therapies have shown to be efficacious in small-animal models, more clinically relevant models are warranted to determine their clinical utility. In an attempt to explore personalized, induced neural stem cell therapy for GBM, Bomba et al. generated, transplanted and investigated the safety of iNSCs in a canine model [68]. Interestingly, pathology findings concluded no signs of abnormal pathology post-mortem and in vitro studies revealed that canine iNSC-sTR maintained their tumoritropic migratory properties and tumor killing capability.

#### 7.1.2. Engineering Stem Cells to Induce Cancer Cell Suicide

SCs can be engineered with suicide genes to express enzymes which convert prodrugs into cytotoxic agents that induce DNA damage in tumor cells, causing tumor cell death. Highly lipophilic anticancer prodrugs are able to penetrate through the BBB to exert tumor killing effects after conversion via the bystander effect. SCs transduced with cytosine deaminase (CD) or thymidine kinase (TK) have been explored as a novel approach to reduce bulk tumor growth while minimizing damage to normal healthy tissue. Cytosine deaminase converts 5-flurocytosine to 5-fluorouracil, a pyrimidine analog. An investigational clinical trial was conducted to assess NSCs expressing E.coli CD in combination with oral 5-flurorcytosine (5-FC) for treatment of recurrent high-grade gliomas [69]. Fifteen patients were enrolled in the study and received intracranial administration of CD-NSCs (10–50 million) followed by oral 5-FC (75–150 mg/kg/day) for 7 days. Results from this study demonstrated that CD-NSCs successfully converted 5-FC to 5-flurouracil (5-FU), indicated by the brain interstitial levels of 5-FU. Furthermore, brain autopsy reports revealed the ability of CD-NSCs to migrate and colocalize with tumor foci in contralateral hemispheres. Although this first-in-human study failed to extend PFS and OS, primary outcome measures for safety and feasibility were achieved. The herpes simplex virus thymidine kinase (HSV-TK) converts ganciclovir (GCV) to (GCV-monohydrate) and is further phosphorylated to GCV-triphosphorylate, which competitively inhibits DNA synthesis in HSV-TK-expressing cells. Several in vivo preclinical studies have shown the clinical feasibility of NSCs and MSCs expressing HSV-TK for the treatment of GBM [70,71,72]. Bomba et al. have also demonstrated the generation and safety of autologous, canine-derived NSCs expressing HSV-TK [68].

#### 7.1.3. Engineering Stem Cells with Oncolytic Virus

Oncolytic viral therapy has been studied in clinical trials for GBM therapy via direct intratumoral injection or within the surgical resection cavity [73,74]. To enhance delivery of virus to tumor, achieve sufficient therapeutic dose, and target distant, invasive tumor foci, SCs have been used as local viral delivery factories. Initial proof-of-concept studies using oncolytic adenovirus with NSCs were conducted to evaluate their transduction efficiency, migration, and intratumoral distribution in vivo [75,76,77,78]. Analysis revealed the ability of NSCs to selectively target, enhance distribution of oncolytic vector, and increase therapeutic efficacy in GBM animal models.

### 7.2. Polymer-Based Scaffolds for Tumoricidal Stem Cells (SCs)

As stated previously, surgical resection remains one of the mainstay treatment options for GBM. Polymeric biodegradable materials have been studied for many years as a solution to locally deliver chemotherapeutics to improve efficacy following surgery [79,80,81]. In particular, hydrogel-based biomaterials have been a growing interest for delivery of therapeutic stem cells. Injectable hydrogels can be prepared using a variety of natural and/or synthetic polymers. These polymers contain functional groups or can undergo surface modifications to facilitate chemical or physical crosslinking in the presence of cells to form an in situ hydrogel following injection. Polymer type, crosslinking method, and concentration of crosslinking linkages are important considerations that can influence hydrogel structural and physical properties (Figure 3) [82]. Natural polymers (such as chitosan, hyaluronic acid, alginate, fibrin collagen, and gelatin) are similar to the native extracellular matrix (ECM), exhibit high biocompatibility, and possess inherent, controllable biodegradability [82]. Alternatively, synthetic polymers (such as poly(ethylene glycol) and poly(lactic acid)) are characterized by their easily tunable, controllable properties [82]. By varying chemical composition and fabrication methods, hydrogel matrix architecture, mechanical strength, and biodegradability can be changed to achieve desired drug/cell release rate. To take advantage of both natural and synthetic properties, hybrid hydrogels have been fabricated to provide suitable scaffold properties for drug/cell delivery [83].

Stimuli-responsive hydrogels or smart hydrogels consist of intelligent polymers that change their physical state, shape, and solvent interactions in response to an external stimulus. This change in transition is reversible when external conditions return to baseline. The driving force that promotes this transition includes a shift in pH and/or temperature. pH or thermosensitive polymers contain functional groups that are highly ionizable (change in net charge) or hydrophobic (lipophilic alkyl moieties) that facilitate alterations in their polymeric structure [84]. The major advantages of smart polymer hydrogel drug/cell delivery systems include reduced frequency of dosing and total dose required, prolonged release of incorporated agent, versatility in route of administration, improved stability and/or protection from drug degradation, and minimized systemic/off-target toxicity.

SCs possess an advantage over conventional chemotherapeutics due to their ability to selectively target, migrate, and kill distant tumor foci. Though intracranial delivery of SCs has been shown to be efficacious in preclinical models, poor SC persistence and retention in the resection cavity limit their clinical utility. The use of biodegradable scaffolds to deliver cytotoxic SCs can prolong their residence time within the resection cavity, which can ultimately enhance anticancer efficacy. Several natural and synthetic polymer-based systems have demonstrated the ability to successfully encapsulate SCs, improve SC persistence, and enhance efficacy of cytotoxic SC therapy for post-surgical GBM [68,85,86,87,88]. Given the advantages of cell-laden polymeric constructs for post-surgical GBM, development of injectable polymeric gels that (1) can form implants in situ in response to physiological temperature and/or pH and (2) can be fine-tuned via crosslinking mechanisms to control biomaterial properties and stem cell release can further increase SC retainability and efficacy.

### 7.3. Immunotherapeutic Strategies for Improving GBM Therapy

As mentioned above, GBM is a highly infiltrative disease, which impedes complete surgical eradication of tumor lesions. The impermeability of the BBB and lack of tumor specificity with conventional chemotherapy reduces efficacy and survival outcomes. The innate and adaptive immune cells provide immunosurveillance by working together to identify and destroy cancer cells. Recent efforts in engineering T cells with chimeric antigen receptors (CAR-T cells) and Fc gamma chimeric receptors (Fcγ-CRs) in addition to the development of therapeutic vaccines using peptide and cell-based platforms have gained considerable traction as promising approaches for providing treatment against tumor specific targets.

#### 7.3.1. Engineering T Cells to Recognize GBM-Associated Antigens and Induce Tumor Cell Death

Adoptive cell therapy (ACT) was first investigated for GBM therapy in the 1980s [89]. ACT is an immunotherapy approach that isolates autologous or allogeneic lymphocytes, expands the lymphocytes ex vivo, and reintroduces them back into the patient to target cancer cells [89]. Although ACT was shown to be safe and demonstrate clinical improvement in some patients, this approach lacks an enrichment of tumor antigen-specific immune cells. These challenges led to the development of CAR-T cells. CAR-T cells are genetically modified T cells that have been engineered to specifically recognize tumor-associated antigens (TAA) that are overexpressed in tumors [90]. The tumor antigen interaction with the CAR construct results in T-cell activation, cytokine release and recruitment of endogenous immune cells, and T-cell proliferation [90]. The CAR construct consists of an extracellular tumor antigen-recognition domain that contains a single-chain variable fragment (scFv) linked to an intracellular T-cell receptor (TCR) signaling domain [89,90,91]. Specific TAAs that have been identified and targeted with CAR-T cell therapy are (1) epidermal growth factor type III variant (EGFRvIII), (2) human epidermal growth factor receptor 2 (HER2), and (3) interleukin-13Rα2 (IL13Rα2) [89,90]. In vivo studies have shown that the use of a first-generation IL13Rα2 CAR-T cell construct was able to target and elicit an effector immune response against glioblastoma cells and GSCs [89]. However, in a phase I clinical trial, the first-generation IL13Rα2 CAR-T cell was unable to elicit an antitumor effector response to eradicate GBM cells [89]. The first-generation CAR-T cell construct contains one T-cell signaling chain, TCR CD3 zeta chain (CD3ζ), in the intracellular domain. Therefore, the poor antitumor activity can be explained by limited T-cell expansion and persistence. To further potentiate the T-cell effector response, second- and third-generation CAR-T cell constructs have been developed. These constructs have been designed with additional co-stimulatory intracellular domains such as CD28, CD134(OX40), and CD137 (4-1BB) that fuse with CD3ζ to boost and sustain the T-cell antitumor activity [89]. One of the biggest challenges with CAR-T cell therapy is cost but more importantly, off-target toxicity concerns associated with rapid release of cytokines (cytokine release syndrome; CRS) that can be life threatening [89]. To mitigate these challenges, Fc gamma chimeric receptor-based (FcγCR) strategies have been employed [89,91]. The structure of FcγCR is very similar to the CAR technology in that it contains a similar intracellular domain; however, the scFv extracellular domain is replaced with the Fc moiety, i.e., CD16 (FcγRIIIA), which is responsible for mediating natural killer (NK) cell antitumor activity [89]. Co-administration of monoclonal antibodies (mAb) with FcγCR offers many advantages over CAR-T technology: (1) the ability to target multiple TAAs with a single FcγCR, and (2) dampen the effects of CRS by discontinuing administration of mAbs [89].

#### 7.3.2. Engineering Vaccines to Stimulate Specific Immune Responses against GBM

The use of therapeutic vaccines is another strategy that has been explored to enhance anticancer immune activity. In 2008, Oncophage, a tumor-derived peptide-based vaccine, was the first vaccine to be granted an orphan drug designation for the treatment of gliomas [92]. Since, many peptide-based and cell-based vaccine strategies have been investigated to overcome the immunosuppressive environment of GBM. These strategies have included the development of dendritic cell-based vaccines to target GSC-specific antigens that are overexpressed in the tumor microenvironment. A randomized phase II clinical trial evaluated the clinical response of dendritic cell vaccines (DCVs) in patients with different molecular expression patterns. Results showed that patients with low levels of B7-H4, a coinhibitory molecule expressed on tumor and tumor-associated macrophages/microglia cells, had significant improvement in overall survival [93]. An early phase clinical trial published in 2018 investigated the effects of peptide vaccine immunotherapy in pediatric patients with low-grade gliomas (LGG) [94]. Results from the clinical trial demonstrated that the peptide vaccine elicited variable immunological response patterns between subjects enrolled in the trial. However, the data were able to show that early elevation of T activation markers was associated with prolonged PFS, in which the data demonstrated an elevation in T-cell activation markers. Further studies to investigate the variability in response patterns should be explored to improve this vaccine platform.

### 7.4. Focused Ultrasound-Mediated Therapy to Improve Delivery of Therapeutics for GBM

Survival of patients with recurrent GBM remains poor and challenging due to the inability of salvage chemotherapeutics to penetrate the BBB (Figure 4A). Focused ultrasound (FUS) in combination with microbubbles is an emerging approach with high potential for effective delivery of therapeutics to the brain. With the proper frequency and pressure of FUS, microbubble oscillation (cavitation) can directly interact with blood vessels endothelium to increase permeability of small molecules and proteins in a reversible fashion (Figure 4B). This has also been shown to induce temporary changes in endothelial cell surface ligands which may increase immune cell extravasation [95]. Using MRI-guided imaging, FUS treatments can precisely be delivered to the entire tumor, thus overcoming the heterogeneous permeability of the BBTB [96]. The physical disruption of the BBB is transient and the barrier functionality and integrity have shown to be completely restored in less than 4–6 h following treatment [96,97,98]. To ensure safety and reproducibility of FUS treatments, studies have investigated the influence of FUS parameters (i.e., frequency, acoustic pressure, pulse repetition frequency (PRF), burst duration, and exposure duration) on FUS-mediated BBB opening. FUS parameters that influence the permeability of the BBB are summarized in Table 3. Additionally, several preclinical studies have shown effective delivery of therapeutic agents to brain tumors using FUS. These studies are summarized in Table 4. Results from several clinical studies have demonstrated the safety and feasibility of using implantable FUS devices and MRI-guided FUS treatments in GBM patients receiving chemotherapy [99,100,101]. Furthermore, ongoing clinical trials assessing the safety of FUS in patients undergoing chemotherapy for primary or recurrent GBM are summarized in Table 5. As stated previously, drugs that are able to traverse the BBB/BBTB are susceptible to drug efflux transporters, thereby reducing drug concentrations in the tumor tissue. Aryal et al. conducted a study to evaluate the effects of p-glycoprotein (Pgp) expression following BBB disruption using FUS and microbubbles [102]. The study was carried out in adult male Sprague-Dawley rats using the following sonication parameters: burst duration 10 ms, frequency 1 Hz, total exposure time 60 s, pressure amplitude 0.55 or 0.81 MPa. The results demonstrated that FUS-induced BBB disruption facilitated by microbubble cavitation can suppress the expression of Pgp. At 0.55 MPa, Pgp expression was suppressed for 48 h, but restored to baseline post-72 h. However, at 0.81 MPa, Pgp expression remained suppressed post-72 h in comparison to baseline. Although this is an initial proof-of-concept study, these results suggest that local inhibition of Pgp using a non-invasive method may enhance retention of drugs in the brain parenchyma and increase drug efficacy.

Interestingly, there is evidence that suggests FUS can mediate the delivery of SCs to the brain. Given that surgery is not clinically feasible for some patients with GBM, the option to intracranially administer therapy may be limited. The novelty in using FUS to systemically administer patient-derived SCs is a promising approach for GBM therapy. In a study conducted by Burgess et al., NSCs were successfully delivered to the brain following BBB disruption with FUS. MRI guidance allowed for specific delivery of NSCs to target site. An alternative study using MSCs explored the underlying molecular mechanisms that may be involved with facilitating cell migration following FUS therapy [103]. The results from this study suggested that endothelial cell surface adhesion molecules are upregulated when stimulated by FUS. This, in turn, enhanced the tumor homing of MSCs 2-fold within the brain tissue.

**Table 3 pharmaceutics-13-01053-t003:** Summary table of focused ultrasound parameters [104].

Parameter	Unit	Definition
Frequency	MHz, Hz	Number of cycles or oscillations per second
Pressure	MPa	Pressure caused by a sound wave minus the ambient pressure in a medium resulting from the sound wave
Pulse repetition frequency	Hz	Number of emitted pulses that occur per second
Burst duration	ms	The length of time designated for repeat pulses at a constant frequency
Total exposure time/total time (TT)	s	The total amount of time the transducer is emitting ultrasonic energy in an area

**Table 4 pharmaceutics-13-01053-t004:** Preclinical and clinical studies using FUS for delivery of therapeutic agents to the brain parenchyma.

Preclinical			
Animal Species/Therapeutic Agent	US Parameters	Key Findings	Ref.
Species: New Zealand white rabbits	Intensity: 16–690 W/cm^2^ Pressure: 0.7–4.7 MPa Burst duration: 10 or 100 ms PRF: 1 Hz TT: 20 s	Low acoustic power levels were able to consistently enhance BBB permeability following administration of an US contrast agent No neuronal damage was observed at pressure amplitudes 0.7 and 1.0 MPa. Opening of the BBB was independent of burst duration and acoustic power.	[96]
Species: Orthotopic xenograft model Drug: Doxorubicin, ado-trastuzumab emtansine (T-DM1)	Frequency: 1 MHz Peak negative pressure (PNP): 480 kPa Burst duration: 10 ms every 1 s TT: 2 min	Extravasation of doxorubicin and T-DM1 was significantly increased using FUS in combination with microbubble contrast agent in comparison to non-FUS group via multiphoton microscopy (7-fold and 2-fold higher). Drug penetration was significantly increased in both treatment groups (>100 vs. <20 μm and 42 ± 7 vs. 12 ± 4 μm for doxorubicin and T-DM1).	[105]
Species: Fischer 344 rats Drug: TMZ	Power: 3 W PNP: 0.6 MPa Burst duration: 10 ms PRF: 1 Hz TT: 60 s	Accumulation of TMZ in CSF/plasma increased following FUS treatment (22.7% to 38.6%). Reduction in 7 day tumor progression ratio was observed following FUS treatment (24.03 to 5.06) Median survival was extended from 20 to 23 following FUS treatment.	[106]
Species: Sprague-Dawley rats Drug: liposomal doxorubicin	Pressure: 1.2 MPa Burst duration: 10 ms PRF: 1 Hz TT: 60–120 s	Reduction in tumor growth was observed in the FUS + DOX treated group in comparison to DOX alone (indicated by tumor volume doubling time 3.7 ± 0.5 days vs. 2.7 ± 0.4 days). A significant increase (>24%) in median survival was observed in FUS + DOX treated group in comparison to non-treated group (*p* = 0.0007).	[107]
Species: Nu/Nu mice Drug: BVZ	Frequency: 400 kHz PNP: 0.4–0.8 MPa Burst duration: 10 ms PRF: 1 Hz TT: 60 s	Penetration of BVZ into the CNS was statistically enhanced in the FUS + BVZ in comparison to BVZ alone (5.73-fold increase at 0.4 MPa and 56.7-fold increase at 0.8 MPa). Median survival time was significantly increased in FUS + BVZ treated group in comparison to BVZ alone (135% vs. 48%; *p* = 0.0002).	[108]

**Table 5 pharmaceutics-13-01053-t005:** Summary of ongoing clinical trials using FUS technology in GBM patients.

NCT Number/Study Completion Date	Status/Location	FUS Device + Drug	Primary Outcome Measures
NCT03616860 Study Completion Date: December 2024	Recruiting Location: Canada	Device: ExAblate Neuro Model 4000 Type 2 Drug: TMZ	Device and procedure related adverse events (safety)
NCT03551249 Study Completion Date: December 2024	Recruiting Location: US	Device: ExAblate Neuro Model 4000 Type 2 Drug: TMZ	Device and procedure related adverse events (safety)
NCT04440358 Study Completion Date: April 2023	Recruiting Location: Canada	Device: ExAblate Neuro Model 4000 Type 2 Drug: Carboplatin	Adverse events (safety) Contrast intensity on MR imaging
NCT04417088 Study Completion Date: November 2023	Recruiting Location: US	Device: ExAblate Neuro Model 4000 Type 2 Drug: Carboplatin	Adverse events (safety) Contrast intensity on MR imaging
NCT03712293 Study Completion Date: December 2021	Recruiting Location: Korea	Device: ExAblate Neuro Model 4000 Type 2 Drug: TMZ	Adverse events (safety)
NCT04446416 Study Completion Date: December 2022	Recruiting Location: Taiwan	Device: NaviFUS System Drug: BVZ	Adverse events (safety) PFS at 6 months

## 8. Conclusions

Is there hope in the future for improving GBM therapy? So far, we know the current standard-of-care treatment for GBM often results in recurrence. However, the management of GBM with immunotherapy in combination with standard therapy may be promising, but more clinical trials are warranted to understand the place of immune checkpoint inhibitors in therapy. Understanding the role of glioma stem cells (GSCs) in mediating chemoresistance and their molecular signatures that drive oncogenic transformation and tumorigenesis is of critical importance for optimizing GBM therapy. In addition, more effort in understanding the structure, physiology, and barrier properties of the BBB/BTB will provide more insight in developing new and/or improving existing technologies to enhance delivery of anticancer therapeutics. Innovative approaches using stem cell therapy, polymeric-based systems, T-cell engineering, therapeutic vaccines, and FUS to improve the delivery of anticancer therapeutics and facilitate drug penetration across the BBB are promising and show benefit in improving GBM treatment. With continued development of these novel approaches, we may see breakthroughs for patients with this devastating, incurable disease.

## Figures and Tables

**Figure 1 pharmaceutics-13-01053-f001:**
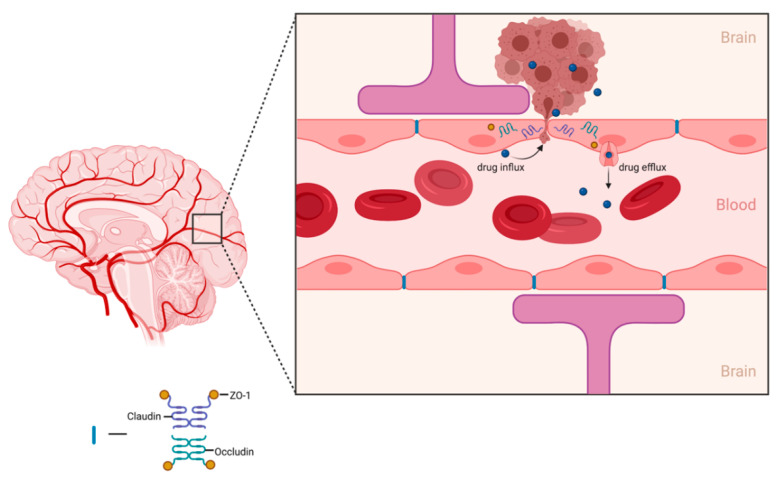
Schematic illustration of drug efflux transporters at the BBB restricting accumulation of drug within the brain parenchyma. Figure created in ©BioRender.

**Figure 2 pharmaceutics-13-01053-f002:**
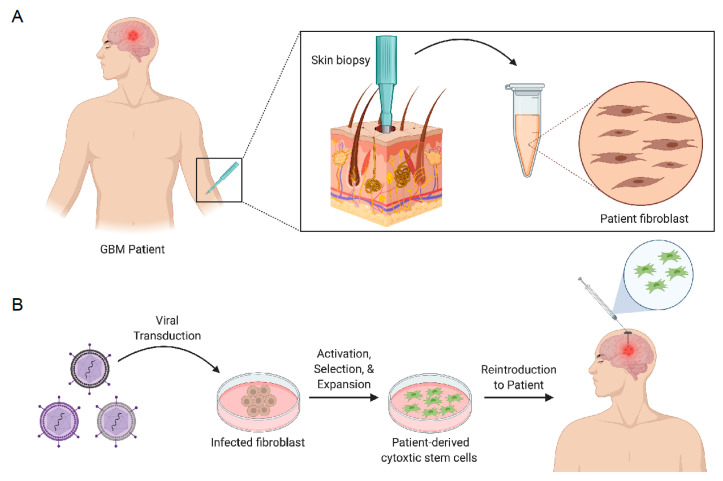
Graphical illustration of skin biopsy obtained from GBM patient (**A**). Viral transduction process of biopsied skin fibroblasts into cytotoxic stem cells for reintroduction into GBM patient (**B**). Figure created in ©BioRender.

**Figure 3 pharmaceutics-13-01053-f003:**
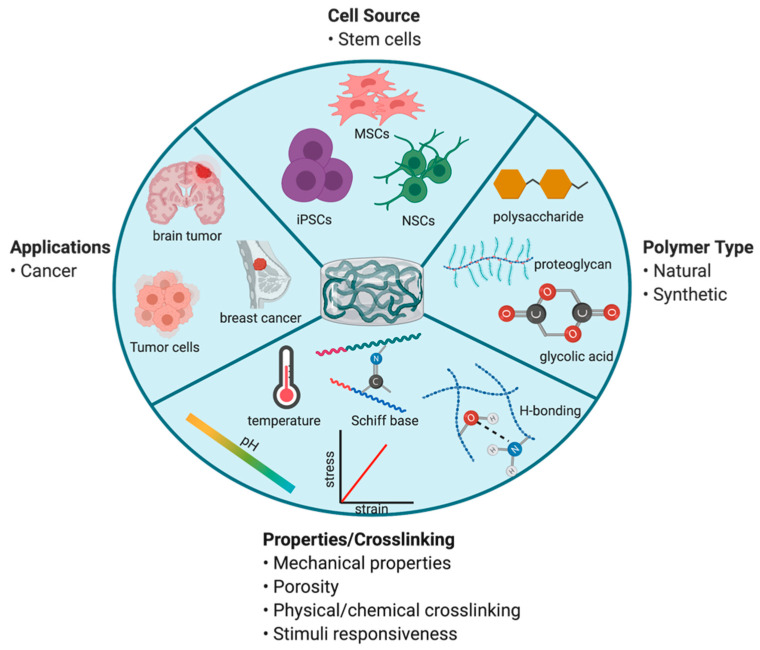
Summary of design and fabrication considerations for injectable cell-laden polymeric gels in cancer applications. Figure created in ©BioRender.

**Figure 4 pharmaceutics-13-01053-f004:**
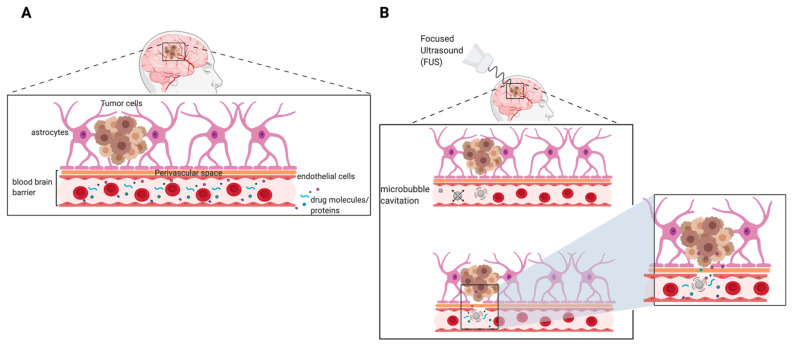
Schematic representation of the inability of drug molecules and proteins to penetrate the BBB in the absence of FUS (**A**). Focused ultrasound-mediated microbubble cavitation facilitates the disruption of the BBB, allowing for enhanced delivery of therapeutic molecules (**B**).

**Table 1 pharmaceutics-13-01053-t001:** Ongoing clinical trials with immune checkpoint inhibitors in GBM.

NCT Number	Official Title	Primary Endpoint(s)	Endpoint Status
NCT02667587	A Randomized Phase 3 Single Blind Study of Temozolomide Plus Radiation Therapy Combined with Nivolumab or Placebo in Newly Diagnosed Adult Subjects with MGMT-Methylated (Tumor-O6-methylguanine DNA Methyltransferase) Glioblastoma (CheckMate-548)	Progression-free survival (PFS) Overall survival (OS)	PFS endpoint not met OS in progress
NCT02337686	Pharmacodynamic Study of Pembrolizumab in Patients with Recurrent Glioblastoma	PFS	Endpoint in progress
NCT03174197	Phase I/II Study to Evaluate the Safety and Clinical Efficacy of Atezolizumab (aPDL1) in Combination with Temozolomide and Radiation in Patients with Newly Diagnosed Glioblastoma	Dose-limiting toxicities (DLT; Phase I) Overall survival (Phase II) Incidence of adverse events	DLT endpoint met OS endpoint not met Study in progress
NCT03047473	Avelumab in Patients with Newly Diagnosed Glioblastoma Multiforme	Safety and tolerability	Endpoint in progress

**Table 2 pharmaceutics-13-01053-t002:** Chemical structures and molecular weights (MW) of small-molecule chemotherapeutics used for the treatment of primary and recurrent GBM.

Chemotherapeutic	Structure (MW)
Temozolomide	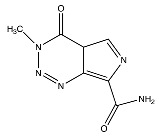 (194.1508 g/mol) ^b^
Lomustine ^a^	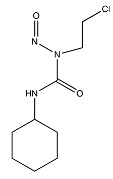 (233.695 g/mol) ^b^
Carmustine ^a^	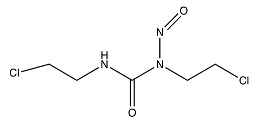 (214.05 g/mol) ^b^
Carboplatin ^a^	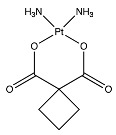 (371.254 g/mol) ^b^

^a^ Drug information provided from public data source of drugbank (https://go.drugbank.com, accessed on 20 December 2020). ^b^ Chemical structures were created using a professional chemical drawing tool (ChemDraw).

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
