# Peer review of "Glioblastoma Multiforme—A Look at the Past and a Glance at the Future"

_pharmaceutics, 2021, doi:10.3390/pharmaceutics13071053_

Round 1

Reviewer 1 Report

The authors in this paper describe the current standard treatments for Glioblastoma and the concomitant adjuvant chemotherapies. I am desolated to not find any description about adoptive cell therapies which are currently ongoing in clinical trials. I strongly recommend the authors insert a paragraph about CAR-T and Fcγ-CRs technologies and currently active CAR-T cell therapies for GBM. In addition, I strongly recommend the authors add the following reviews on this topic:

Caratelli, S. et al. FCgamma Chimeric Receptor-Engineered T Cells: Methodology, Advantages, Limitations, and Clinical Relevance. Frontiers in immunology 8, 457, doi:10.3389/fimmu.2017.00457 (2017).

Marei HE, et al. Recent perspective on CAR and Fcγ-CR T cell immunotherapy for cancers: Preclinical evidence versus clinical outcomes. Biochemical pharmacology. 2019;166:335-46.

Several clinical trials are ongoing with the aim to assess the safety and efficacy of CAR T cells for GBM using different GBM-associated antigens such as EGFRvIII, NKG2D, B7-H3, CD147, IL13Ralpha2, and HER2.

I think that the manuscript is fairly acceptable for publication on Pharmaceutics, provide that the authors would answer the above concerns and implement the manuscript accordingly.

Author Response

Thank you for your valued feedback and recommendations. We have provided a point-by-point response in the attached document to address the reviewer's comments to the best of our ability. 

  1. The authors in this paper describe the current standard treatments for Glioblastoma and the concomitant adjuvant chemotherapies. I am desolated to not find any description about adoptive cell therapies which are currently ongoingin clinical trials. I strongly recommend the authors insert a paragraph about CAR-T and Fc?-CRs technologies and currently active CAR-T cell therapies for GBM. In addition, I strongly recommend the authors add the following reviews on this topic:

Caratelli, S. et al. FCgamma Chimeric Receptor-Engineered T Cells: Methodology, Advantages, Limitations, and Clinical Relevance. Frontiers in immunology 8, 457, doi:10.3389/fimmu.2017.00457 (2017).

 Marei HE, et al. Recent perspective on CAR and Fcγ-CR T cell immunotherapy for cancers: Preclinical evidence versus clinical outcomes. Biochemical pharmacology. 2019;166:335-46.

Response: Thank you for this recommendation.We fully agree that current cell-based technologies should be included in this review. We have addressed this in Chapter 7 of the revised review entitled “Immunotherapeutic strategies for improving GBM therapy”.We have also included the reviews from Caraterlli et al. and Marei et al. as recommended.

Reviewer 2 Report

King and Benhabbour presented a well-structured review article on the therapeutic options available for the treatment of glioblastoma. The authors provided a comprehensive description of the past and novel therapeutic approaches available highlighting strengths and limitations. Overall, the manuscript appears complete and well written, however, there is some missing information that could improve the novelty of the present manuscript. Please, see the following comments:
1) Please provide references for the following sentences: “GBM is the most aggressive, highly malignant tumor of the astrocytic lineage and is commonly diagnosed in elderly patients (median age at diagnosis ≥ 65 years). Glioblastomas are characterized by extensive, diffuse tumor invasion and infiltration, microvascular proliferation, and high genomic instability.”. In addition, data about the molecular subtypes of GBM and the consequent therapeutic options should be added. For this purpose, see:
- PMID: 31405017
- PMID: 24114756
- PMID: 27697594
- PMID: 25084005
2) Throughout the manuscript the authors state twice “...GBM remains incurable and is universally fatal....”. Although difficult to treat, glioblastoma is not always lethal so authors should find a less strong definition than the one given;
3) Please provide supporting references (if available) for the following paragraph: “These studies are currently awaiting publication but, preliminary data have been reported. In the CheckMate-548 clinical trial, the combination of nivolumab with first-line GBM therapy failed to meet PFS primary outcome measure and the investigators are currently awaiting for overall survival data (NCT02667587). Similarly, disappointing primary outcome measures were observed in the phase II study evaluating the safety and efficacy of atezolizumab in combination with first-line treatment (NCT03174197). However, reports showed that concurrent atezolizumab with TMZ and radiotherapy was well tolerated and no safety concerns were observed. GBM recurrence was observed in several patients post atezolizumab treatment. Seventeen patients received repeat surgery and analysis of tumor tissue pre- and post-immunotherapy may provide clinical insight on immune-therapy resistance. A phase II study is evaluating the pharmacodynamic effects of pembrolizumab in newly diagnosed patients with GBM. Twenty participants have been enrolled in this study and currently there are no study updates on primary outcome measures (NCT02337686). Lastly, in a single center, phase 2, open label study the combi- 190 nation of avelumab with standard therapy has shown to be safe and generally well toler- 191 ated in newly diagnosed GBM patients (NCT03047473). Although the efficacy data is premature and results are preliminary, avelumab may be promising at the initial stages of GBM therapy.”;
4) In chapter 7 the authors have to add some information about the use of therapeutic anticancer vaccine mainly using oncophages as well as some information about CAR-T cell therapy for glioblastoma treatment. For this purpose, please see:
- PMID: 31575023 
- PMID: 29618666
- PMID: 30483135
- PMID: 30159779
- PMID: 29509936

Author Response

Thank you for your valued feedback and recommendations. We have provided a point-by-point response in the attached document to address the reviewer's comments to the best of our ability. 

King and Benhabbour presented a well-structured review article on the therapeutic options available for the treatment of glioblastoma. The authors provided a comprehensive description of the past and novel therapeutic approaches available highlighting strengths and limitations. Overall, the manuscript appears complete and well written, however, there is some missing information that could improve the novelty of the present manuscript. Please, see the following comments:

  1. Please provide references for the following sentences: “GBM is the most aggressive, highly malignant tumor of the astrocytic lineage and is commonly diagnosed in elderly patients (median age at diagnosis ≥ 65 years). Glioblastomas are characterized by extensive, diffuse tumor invasion and infiltration, microvascular proliferation, and high genomic instability.”. In addition, data about the molecular subtypes of GBM and the consequent therapeutic options should be added.

 Response: Thank you for this comment. References have been addedfor the sentences above and we have included information about molecular subtypes in GBM. This information can be found in the Chapter 1 –“Introduction” sectionof the revised review.

  1. Throughout the manuscript the authors state twice “...GBM remains incurable and is universally fatal....”. Although difficult to treat, glioblastoma is not always lethal so authors should find a less strong definition than the one given.

Response: Thank you for this recommendation. We have made some changes to this statement. Thesechanges can be reviewed in Chapter 1 –“Introduction” section

  1. Please provide supporting references (if available) for the following paragraph: “These studies are currently awaiting publication but, preliminary data have been reported. In the CheckMate-548 clinical trial, the combination of nivolumab with first-line GBM therapy failed to meet PFS primary outcome measure and the investigators are currently awaiting for overall survival data (NCT02667587). Similarly, disappointing primary outcome measures were observed in the phase II study evaluating the safety and efficacy of atezolizumab in combination with first-line treatment (NCT03174197). However, reports showed that concurrent atezolizumab with TMZ and radiotherapy was well tolerated and no safety concerns were observed. GBM recurrence was observed in several patients post atezolizumab treatment. Seventeen patients received repeat surgery and analysis of tumor tissue pre-and post-immunotherapy may provide clinical insight on immune-therapy resistance. A phase II study is evaluating the pharmacodynamic effects of pembrolizumab in newly diagnosed patients with GBM. Twenty participants have been enrolled in this study and currently there are no study updates on primary outcome measures (NCT02337686). Lastly, in a single center, phase 2, open label study the combi-190 nation of avelumab with standard therapy has shown to be safe and generally well toler-191 ated in newlydiagnosed GBM patients (NCT03047473). Although the efficacy data is premature and results are preliminary, avelumab may be promising at the initial stages of GBM therapy.”;

Response: Thank you forthis comment. Publications and referencesof the clinicaltrials listed above have notyetbeen made available for open access. As such, we did not make any changes to the above paragraph.

  1. In chapter 7 the authors have toadd some information about the use of therapeutic anticancer vaccine mainly using oncophages as well as some information about CAR-T cell therapy for glioblastoma treatment.

Response: Thank youfor this recommendation.We fully agree this information should be reflected in this review. We have addressed this in Chapter 7 of the review and added a section entitled “Immunotherapeutic strategies for improving GBM therapy”.